The evolutionary history and diagnostic utility of the CRISPR-Cas system within Salmonella enterica ssp. enterica

Pettengill James B. 1 james.pettengill@fda.hhs.gov
Timme Ruth E. 1
Barrangou Rodolphe 2
Toro Magaly 3
Allard Marc W. 1
Strain Errol 1
Musser Steven M. 1
Brown Eric W. 1
1 Center for Food Safety & Applied Nutrition, US Food & Drug Administration , College Park, MD , USA
2 Department of Food, Bioprocessing and Nutrition Sciences, North Carolina State University , Raleigh, NC , USA
3 Department of Nutrition and Food Science, University of Maryland , College Park, MD , USA
Wiles Siouxsie
Electronic publication date: 2014 Apr 17
Publication date: 2014
Volume: 2
Electronic Location ID: e340
Received 2014 Jan 3; Accepted 2014 Mar 21
Copyright: © 2014 Pettengill et al.
Copyright year: 2014
Copyright holder: Pettengill et al.
License: This is an open access article distributed under the terms of the Creative Commons Attribution License, which permits unrestricted use, distribution, and reproduction in any medium, provided the original author and source are credited.
License URL: https://creativecommons.org/licenses/by/3.0/

Keywords: Salmonella, Horizontal gene transfer, Evolution, CRISPR, Outbreak, Phylogeny, Whole genome sequencing, Typing

Funding: This work was supported by the Center for Food Safety and Applied Nutrition at the US Food and Drug Administration. The funders had no role in study design, data collection and analysis, decision to publish, or preparation of the manuscript.

==============================
Evolutionary studies of clustered regularly interspaced short palindromic repeats (CRISPRs) and their associated (cas) genes can provide insights into host-pathogen co-evolutionary dynamics and the frequency at which different genomic events (e.g., horizontal vs. vertical transmission) occur. Within this study, we used whole genome sequence (WGS) data to determine the evolutionary history and genetic diversity of CRISPR loci and cas genes among a diverse set of 427 Salmonella enterica ssp. enterica isolates representing 64 different serovars. We also evaluated the performance of CRISPR loci for typing when compared to whole genome and multilocus sequence typing (MLST) approaches. We found that there was high diversity in array length within both CRISPR1 (median = 22; min = 3; max = 79) and CRISPR2 (median = 27; min = 2; max = 221). There was also much diversity within serovars (e.g., arrays differed by as many as 50 repeat-spacer units among Salmonella ser. Senftenberg isolates). Interestingly, we found that there are two general cas gene profiles that do not track phylogenetic relationships, which suggests that non-vertical transmission events have occurred frequently throughout the evolutionary history of the sampled isolates. There is also considerable variation among the ranges of pairwise distances estimated within each cas gene, which may be indicative of the strength of natural selection acting on those genes. We developed a novel clustering approach based on CRISPR spacer content, but found that typing based on CRISPRs was less accurate than the MLST-based alternative; typing based on WGS data was the most accurate. Notwithstanding cost and accessibility, we anticipate that draft genome sequencing, due to its greater discriminatory power, will eventually become routine for traceback investigations.

Introduction

Clustered regularly interspaced short palindromic repeats (CRISPRs) represent a unique and peculiar genomic element within many Archaeal and Bacterial groups (Barrangou, 2013; Haft et al., 2005; Horvath & Barrangou, 2010). They are formed through the acquisition of exogenous nucleic acids (termed spacers) that are embedded between endogenous DNA sequences (termed repeats, that are usually 21 to 47 bp). The spacers may be acquired from a number of different sources including phages or plasmids (Barrangou et al., 2007; Westra et al., 2012). Construction and maintenance of the CRISPR array occurs through the CRISPR-associated (Cas) proteins, which identify foreign phage nucleic elements (termed proto-spacers) and incorporate them into the CRISPR locus (Marraffini & Sontheimer, 2010). Spacers are transcribed and processed into small interfering RNAs that guide the Cas machinery towards complementary nucleic acids for sequence-specific cleavage (Bhaya, Davison & Barrangou, 2011). The proposed biological significance of the CRISPR system is that through the incorporation of spacer elements, which serves as a monitoring system, the bacterium gains some degree of immunity to the harmful foreign elements within its environment (Barrangou et al., 2007; Horvath & Barrangou, 2010; van der Oost et al., 2009).

In addition to investigating the cellular and molecular details of how the CRISPR-Cas system functions (e.g., Barrangou et al., 2007; Beloglazova et al., 2011; Karginov & Hannon, 2010), a full understanding of the system requires macro-evolutionary studies. Such studies can provide important information on host-pathogen co-evolutionary dynamics (England & Whitaker, 2013) and transmission (e.g., vertical or horizontal) rates of a system that may substantially alter an individual’s fitness (Jiang et al., 2013; Levin, 2010). Three main CRISPR-Cas system have been identified (Type I, Type II, Type III; (Makarova, Wolf & Koonin, 2013)) and studies have found an appreciable amount of diversity at higher taxonomic levels (e.g., phylum) in the makeup of these systems. What is somewhat surprising is that the distribution of those systems across a phylogeny based on 205 Cas1 sequences representing 2,262 genomes shows a high degree of polyphyly and multiple instances of independent evolution (Makarova, Wolf & Koonin, 2013), which is likely the result of horizontal transmission events. The pattern of polyphyletic types and the lack of phylogenetic congruence between cas genes and that of the genomic background, assumed to be vertically inherited, also extends to lower taxonomic ranks (e.g., among Salmonella serovars Timme et al., 2013; Touchon & Rocha, 2010). There is also evidence for significant variation within serovars as to the CRISPR array length and the presence absence of cas genes (Cain & Boinett, 2013; Makarova, Wolf & Koonin, 2013).

In addition to understanding CRISPRs from an evolutionary perspective, their utility as a marker for serotyping and subtyping has been investigated (e.g., Fabre et al., 2012; Liu et al., 2011). There are two primary characteristics of CRISPRs that make them suitable for such a purpose. First, due to the evolutionary arms race between foreign elements (e.g., phages) and the host bacteria, the associated rapidly changing selection pressures may cause CRISPRs to evolve quite quickly (Karginov & Hannon, 2010; Tyson & Banfield, 2008). Thus, through the acquisition of new and deletion of old spacers, differences useful for typing and subtyping (in Salmonella, this refers to strain identification and differentiation at the serovar and sub-serovar level) could arise even between closely related strains. Second, the spacer regions serve as a DNA fingerprint that might characterize the source environment. Noteworthy is the polarized addition of novel spacers at one end of the repeat-spacer array, which provides insights into sequential events that occurred over time. As a result, spacers may represent a biogeographic marker that could be useful for differentiating individuals that are found in different environments (e.g., Candidatus Accumulibacter phosphatis Kunin et al., 2008 and Sulfolobus islandicus Held & Whitaker, 2009). These properties of CRISPRs, which may lead to high levels of polymorphism, have proven useful to characterize Yersinia pestis (Pourcel, Salvignol & Vergnaud, 2005), Campylobacter jejuni (Schouls et al., 2003), and to subtype Mycobacterium tuberculosis associated with an outbreak investigation (Groenen et al., 1993).

The utility of CRISPRs as a subtyping tool within Salmonella has also been explored (Fabre et al., 2012; Liu et al., 2011), which is not surprising as Salmonella enterica subsp. enterica is the leading cause of bacterial food-borne disease in the United States (CDC, 2011; Voetsch et al., 2004) and any potentially useful molecular markers are of great interest to the public health community. Although recombination and horizontal gene transfer has likely occurred throughout the macro-evolutionary history of Salmonella (Brown et al., 2003; Octavia & Lan, 2006), within serovar lineages are often highly clonal (den Bakker et al., 2011a; Le et al., 2007; Zhou et al., 2013). Traditional typing methods such as pulsed-field gel electrophoresis (PFGE), multilocus sequence typing (MLST), or multiple-locus variable-number tandem repeat analysis (MLVA) often can not differentiate among the highly clonal isolates within serovars (e.g., Leekitcharoenphon et al., 2014; Malachowa et al., 2005; Perez-Losada et al., 2013). This necessitates the development of markers with higher discriminatory power and CRISPRs, with their hypervariable spacer content, may represent such a marker.

Within Salmonella, which contains the Type I-E CRISPR-Cas system (Makarova et al., 2011), there are two CRISPR loci (CRISPR1 and CRISPR2) that differ in both the identity and number of spacers and repeats (Jansen et al., 2002; Touchon & Rocha, 2010). Previous studies have arrived at contradicting conclusions regarding the utility of these genomic regions as a typing or subtyping tool. For example, based on a study of 130 serovars Fabre et al. (2012) found that CRISPR polymorphism correlated strongly with serovar. Hence, they concluded that they should be useful for strain tracking and developed a high-throughput subtyping assay for Salmonella Typhimirium (Fabre et al., 2012). In contrast, Touchon & Rocha (2010) concluded that CRISPR loci within enterobacteria are likely to be poor epidemiological markers given the slow rate at which they evolve and the lack of congruence among the cas genes, CRISPR loci and species phylogeny. However, it is important to note that Touchon & Rocha (2010) focused primarily on a subset of E. coli strains. Timme et al. (2013) also reported incongruence in Salmonella between evolutionary patterns based on CRISPR loci and phylogenetic relationships inferred using whole genome sequence data, although that study focused on between-serovar differences and did not investigate the subtyping capability of CRISPR loci.

In this study, we analyzed CRISPR loci identified from 427 whole genome sequences representing 64 different serovars of Salmonella enterica ssp. enterica. First, we described the patterns of CRISPR and cas diversity across this diverse set of isolates and investigated their evolutionary history through phylogenetic reconstruction. Next, we evaluated the performance of whole genome sequence data, MLST and CRISPRs for typing isolates based on how often the clusters were congruent with taxonomic groups (i.e., did the different methods reconstruct monophyletic groups). For typing with CRISPRs, we developed and describe a novel approach employing a model-based Bayesian method to cluster isolates based on CRISPR spacer similarity.

Results

CRISPR and cas gene diversity

Across the 431 isolates we observed 878 unique spacers and 75 unique repeats within CRISPR1. For CRISPR2, we found 1,241 unique spacers and 65 unique repeats. The average length of CRISPR1 and CRISPR2 was 14 and 17 repeat units, respectively (Table 1; Fig. 1). The extreme length of CRISPR2 (221 units) within the Mbandaka isolate in part drives the difference in average length between the two arrays (Table 1; Fig. 1). There was also a great deal of length variability within many of the serovars (Fig. 1). For example, S. Senftenberg isolates had CRISPR1 array lengths that differed by as many as 50 units. Virchow also had an appreciable level of intra-serovar diversity particularly within CRISPR2. Although we had a heavily skewed sampling scheme where the vast majority of serovars had less than three isolates, this does not account for the differences in array length given that some serovars with only a few isolates (e.g., Senftenberg and Muenster; Fig. 1) had greater variation in array length than those serovars for which we had many isolates. Spacer diversity within either CRISPR locus did not differ between the two major clades of S. enterica previously identified in den Bakker et al. (2011b) and Timme et al. (2013) (e.g., Clades A and B; Table 1).

Figure 1 CRISPR variation.

Variation in CRISPR locus length among the 431 isolates for both locus 1 and locus 2. Boxes depict the interquartile (IQR) range and whiskers indicate 1.5 IQR; the horizontal black line represents the mean.

Table 1 The major clade within S. enterica to which each serovar belongs, number of isolates (N), and the average length of CRISPR1 (L1) and CRISPR2 (L2). Lengths are the number of spacers.

Subspecies and Serovar	Clade	N	L 1	L 2	
S. subsp. enterica ser. Abony	A	2	14	4	
S. subsp. enterica ser. Agona	A	37	18	35	
S. subsp. enterica ser. Albany	A	1	17	16	
S. subsp. enterica ser. Anatum	A	2	47	10	
S. subsp. enterica ser. Bareilly	A	2	13	27	
S. subsp. enterica ser. Berta	A	1	11	6	
S. subsp. enterica ser. Braenderup	A	2	41	55	
S. subsp. enterica ser. Bredeney	B	1	18	24	
S. subsp. enterica ser. Cerro	A	1	43	48	
S. subsp. enterica ser. Chester	B	1	3	2	
S. subsp. enterica ser. Choleraesuis	A	2	16	10	
S. subsp. enterica ser. Cubana	A	1	25	28	
S. subsp. enterica ser. Derby	A	1	27	60	
S. subsp. enterica ser. Dublin	A	2	11	5	
S. subsp. enterica ser. Eastbourne	B	1	3	2	
S. subsp. enterica ser. Enteritidis	A	103	19	18	
S. subsp. enterica ser. Galinarum	A	1	21	4	
S. subsp. enterica ser. Gaminara	B	1	32	38	
S. subsp. enterica ser. Give var15-34	B	1	70	26	
S. subsp. enterica ser. Hadar	A	1	32	56	
S. subsp. enterica ser. Hartford	A	1	32	51	
S. subsp. enterica ser. Havana	A	1	3	40	
S. subsp. enterica ser. Heidelberg	A	55	33	52	
S. subsp. enterica ser. Indiana	A	1	35	44	
S. subsp. enterica ser. Inverness	A	1	21	20	
S. subsp. enterica ser. Javiana	B	3	22	15	
S. subsp. enterica ser. Kentucky	A	9	42	30	
S. subsp. enterica ser. Kunzendorf	A	1	15	8	
S. subsp. enterica ser. Litchfield	A	1	47	4	
S. subsp. enterica ser. London	A	1	29	35	
S. subsp. enterica ser. Manhattan	A	1	5	22	
S. subsp. enterica ser. Mbandaka	A	1	33	221	
S. subsp. enterica ser. Meleagridis	A	1	51	48	
S. subsp. enterica ser. Miami	B	1	15	16	
S. subsp. enterica ser. Minnesota	B	1	39	26	
S. subsp. enterica ser. Montevideo	B	51	33	38	
S. subsp. enterica ser. Muenchen	A	3	14	31	
S. subsp. enterica ser. Muenster	B	2	48	126	
S. subsp. enterica ser. Nchanga	A	2	20	10	
S. subsp. enterica ser. Newport	A	61	32	27	
S. subsp. enterica ser. Norwich	A	1	5	4	
S. subsp. enterica ser. Ohio	A	1	7	5	
S. subsp. enterica ser. Oranienburg	B	3	19	56	
S. subsp. enterica ser. Panama	B	1	19	2	
S. subsp. enterica ser. Paratyphi A	A	1	7	14	
S. subsp. enterica ser. Paratyphi B	A	8	19	28	
S. subsp. enterica ser. Pomona	B	1	23	2	
S. subsp. enterica ser. Poona	B	1	11	72	
S. subsp. enterica ser. Pullorum	A	4	13	4	
S. subsp. enterica ser. Reading	B	1	69	77	
S. subsp. enterica ser. Rissen	A	1	47	60	
S. subsp. enterica ser. Rubislaw	B	1	3	2	
S. subsp. enterica ser. Saintpaul	A	3	44	23	
S. subsp. enterica ser. Senftenberg	A	8	61	51	
S. subsp. enterica ser. Sloterdijk	A	1	29	42	
S. subsp. enterica ser. Soerenga	A	1	79	70	
S. subsp. enterica ser. Stanley	A	1	41	22	
S. subsp. enterica ser. Stanleyville	A	1	21	4	
S. subsp. enterica ser. Tallahassee	A	1	17	6	
S. subsp. enterica ser. Tennessee	A	4	40	66	
S. subsp. enterica ser. Typhimurium	A	18	53	43	
S. subsp. enterica ser. Urbana	B	1	23	4	
S. subsp. enterica ser. Virchow	A	1	33	72	
S. subsp. enterica ser. Worthington	A	1	16	21	
S. subsp. houtenae ser. 50:g,z51:- str. 01-0133	N/A	1	15	40	
S. subsp. salamae ser. 58:l,z13,z28:z6 str. 00-0163	N/A	1	29	73	
S. subsp. diarizonae ser. 60:r:e,n,x,z15 str. 01-0170	N/A	1	17	0	
S. subsp. indica ser. 6,14,25:z10:1,(2),7 str. 1121	N/A	1	3	40	
Summary*	N/A	431	27	33	
Notes.

* The total number of isolates and average lengths of CRISPR1 and CRISPR2 across all serovars.

For 325 of the 431 isolates, all eight cas genes (Type I-E CRISPR-Cas system) were present in the genome (cas3, cse1, cse2, cas2, cas7, cas5, cas6e, cas1) (Table S1). The five-prime most gene, cas3 was missing in 69 genomes. Noteworthy, in 20 other genomes it was the only cas gene present. Twenty-seven genomes had at least one additional cas gene missing, and 10 had a complete absence of any cas gene. There were no apparent differences between serovars within Clade A and B in the presence/absence of cas genes (Table S1).

Based on a cas gene tree reconstructed from a concatenation of all eight cas genes, there are two general sequence profiles present across the 431 isolates, which we refer to as cas type a (I-Ea) and cas type b (I-Eb) (Fig. 2). The cas gene tree is incongruent with respect to the whole-genome SNP phylogeny (Fig. 2) suggesting a non-vertical transmission mechanism for the observed cas gene sequence types.

Figure 2 Whole genome and cas gene phylogenies.

Phylogenetic relationships among the 431 isolates determined using whole genome sequencing data from which a SNP matrix was created using the k-mer based approach implemented in kSNP [39]. Bootstrap values are based on 100 traditional replicates created using seqboot within the phylip package [60]. The two cas gene profiles are also shown as different colors at the tips (cas type a (I-Ea) = blue; cas type b (I-Eb) = red). Branch width is indicative of bootstrap support value (thickest lines depict >80% bootstrap support). Gray colored branches represent lineages found in Clade B [16,38]; all other lineages except the outgroups belong to Clade A. The insert shows the phylogenetic relationships based on a phylogeny constructed using only the cas genes with tips colored according to cas type as shown in the larger phylogeny.

Analyses of the cas genes individually also reflects the strong differentiation into two groups given the predominantly bimodal distribution in pairwise distances within each gene (Fig. S1). However, there are differences among the genes in the range of distances between isolates. For example, cas1 and cas2 have the smallest range of pairwise distances with cse2 having the largest, which may provide some insight into the selective constraints acting on the loci (e.g., purifying selection is greatest on cas1 and cas2).

Typing and subtyping

The SNP matrix created using the program kSNP (Gardner & Slezak, 2010) had 207,797 nucleotides. Phylogenetic reconstruction based on this SNP matrix resolved the two major clades of Salmonella enterica ssp. enterica and other relationships (Fig. 2) that have been observed elsewhere (den Bakker et al., 2011b; Timme et al., 2013). The gsi (genealogical sorting index) values, which provide a measure of how well the topology based on the SNP matrix reconstructed relationships consistent with taxonomic categories (i.e., strains from the same serovar form a reciprocally monophyletic group), was 1.0 for 14 of the 23 serovars for which we had greater than one isolate (Table S2, Fig. 3).

Figure 3 Genealogical sorting index (gsi) results per dataset.

Boxplot illustrating the differences among the four datasets in gsi values, which was used as a metric to quantify how well the datasets constructed relationships congruent with taxonomy. Boxes depict the interquartile (IQR) range and whiskers indicate 1.5 IQR; the horizontal black line represents the mean. Gray dots represent observed values within each dataset and are dispersed horizontally (jittered) to decrease overlap.

Within the MLST dataset, we observed 3,345 total nucleotides of which 453 were variable. As expected given the smaller number of variable sites, topological relationships were not as well supported as what was observed on the SNP tree; there are also differences between the two datasets in the evolutionary relationships inferred (Fig. 4). In particular, the general differentiation of isolates into Clade A and Clade B observed in the SNP tree and within other studies (den Bakker et al., 2011b; Timme et al., 2013) was not observed, and Montevideo is found among serovars that typically define Clade A rather than Clade B. One note of congruence between the MLST and whole genome phylogeny is the presence of two Newport lineages (Figs. 2 and 4). However, as noted above, there is little topological support for basal relationships inferred with the MLST data. A gsi value of 1.0 was observed for 11 of the 23 serovars with multiple isolates (Table S2, Fig. 3).

Figure 4 MLST phylogeny.

Phylogenetic relationships among the sampled isolates based on MLST matrix. Branch width is indicative of bootstrap support value (thickest lines depict >80% bootstrap support).

We used two approaches to determine how well the CRISPR loci could be used to type isolates. For the first, we used the program uclust (Edgar, 2010) to create groups of like spacer sequences and then constructed a topology based on a binary matrix created by determining the presence/absence of each isolate within those groups. For CRISPR1, we observed 824 clusters within average size of 11.6 spacers; for CRISPR2, we found 1,176 clusters with an average size of 11.9 spacers. The phenograms based on the spacers for each CRISPR1 and CRISPR2 had marginally better support than the MLST dataset. The topologies inferred from the CRISPR loci differed with the MLST and the SNP trees (Figs. 5 and 6). However, this result is not really unexpected as it is unlikely that spacer content would necessarily reflect phylogenetic relationships, as they are not always vertically transmitted (Horvath & Barrangou, 2010; Touchon & Rocha, 2010). Focusing on clustering patterns of isolates from the same serovar, we found that they formed monophyletic groups most often under the SNP dataset followed by the MLST and CRISPR loci (under both CRISPR loci only 9 serovars had gsi values of 1; Table S2, Figs. 2–6). There are also some odd clustering patterns based on the spacer content of CRISPRs. For example, the phenogram for both CRISPR loci show a Munchen isolate embedded within one of the Newport groups (Figs. 5 and 6). Also peculiar is that the Newport isolates are found within three clusters based on spacer content within CRISPR2; they are found in two clusters within the three other datasets.

Figure 5 CRISPR1 phenogram.

Phenogram depicting similarity among isolates in spacer composition of CRISPR1. Branch width is indicative of bootstrap support value (thickest lines depict >80% bootstrap support).

Figure 6 CRISPR2 phenogram.

Phenogram depicting similarity among isolates in spacer composition of CRISPR2. Branch width is indicative of bootstrap support value (thickest lines depict >80% bootstrap support).

The second approach we used was the model-based Bayesian clustering algorithm implemented in the program STRUCTURE (Falush, Stephens & Pritchard, 2003), which incorporates no a priori information when assigning individuals to clusters. Rather, the method groups individuals based on the similarity deduced from a presence/absence matrix of individuals within different clusters created based on spacer similarity (see Materials and Methods). In general, we observed patterns similar to the phylogenetic analysis in that the majority of isolates from the same serovar are clustered together and the Newport individuals break out into two distinct groups (Fig. 7). However, compared to the phylogenetic and clustering analyses we were also able to determine the degree to which individuals from different serovars have some spacers in common. For example, the pattern within Saintpaul suggests that those isolates not only have spacers within both CRISPR1 and CRISPR2 that are unique but also spacers that resemble those found in Paratyphi B isolates; the Senftenberg and Javiana isolates also appear to have spacers found in other isolates (Fig. 7).

Figure 7 DISTRUCT diagram depicting clusters based on CRISPR spacer similarity.

Model-based clustering results showing the assignment of individuals to different groups based on similarity in SNP profiles. Only serovars for which >3 isolates were sequenced are shown. Colors indicate the different clusters and the degree to which a vertical bar consists of multiple colors is indicative of the proportion of SNPs that resemble a particular cluster.

Intra- and inter-serovar pairwise distances

We also compared estimates of pairwise distances among isolates from the same serovar to pairwise distances among isolates from different serovars. Such information can provide additional insight into the utility of each marker for typing or subtyping since if there is a great deal of overlap between the distance classes then such a marker will not be useful. Using this approach, we found that the SNP matrix had the largest gap between the distance classes in that there were virtually no inter-serovar pairwise comparisons that were of a similar small magnitude as the intra-specific comparisons (Fig. 8, Table 2). However, there are exceptions where isolates from different serovars have a pairwise distance on par with what is expected for isolates from the same serovar (e.g., a Senftenberg isolate and Tennessee isolate) due to high sequence similarity. There are also instances where intra-serovar pairwise distances are similar to the magnitude observed between serovars, which is primarily the result of isolates within Paratyphi B, Newport, and Kentucky, which is not surprising as these serovars are polyphyletic within the SNP phylogeny (Fig. 2). A similar pattern is also observed for the MLST dataset but there are more inter-serovar comparisons that are of the magnitude observed among within serovar comparisons.

Figure 8 Pairwise distances among individuals for each dataset.

Intra- and inter-serovar pairwise distance histograms for (A) the kSNP matrix, (B) MLST matrix, (C) CRISPR1 presence/absence matrix, and (D) CRISPR2 presence/absence matrix. Note that scales on the x-axis differ due to the method used to calculate distances and scale on the y-axis differs as a result of different binwidths and distribution of observations within each bin.

Table 2 Mean intra- and inter-serovar pairwise distance estimates for the four different marker datasets.

Marker type	Intraspecific	Interspecific	
CRISPR1	1.9243	4.9884	
CRISPR2	1.4003	5.4235	
SNP	0.0068	0.0725	
MLST	0.0011	0.0112	

For the CRISPR loci, both exhibited the expected pattern of inter-serovar distances being substantially larger than intra-serovar comparisons (Fig. 8, Table 2). However, in contrast to the MLST and SNP datasets, there are many intra-serovar pairwise comparisons that are of a similar magnitude to inter-serovar pairwise distances. This is not surprising given that there is a diversity of spacers within each serovar, which would result in individual isolates from the same serovar being assigned to different clusters.

Discussion

The CRISPR-Cas system and the putative immunity it provides for bacteria represents a significant discovery within microbiology and evolutionary biology in general. The research avenues created by this discovery are numerous. Within this study we focused on the CRISPR-Cas system within Salmonella enterica ssp. enterica, providing insights into both the history of this system and evaluation of its utility for typing isolates. We found two distinct cas gene profiles that are not congruent with phylogenetic relationships suggesting that horizontal transmission events are responsible. Based on the clustering method implemented in this study that captures differences in spacer content, we found that the CRISPR loci may contain sufficient information to be useful in typing certain isolates. However, the degree of false positives (i.e., the topological placement of an isolate within a serovar group to which it did not belong) was higher than that observed when typing based on MLST loci. Both MLST and CRISPRs performed poorly relative to clustering results based on SNPs mined from WGS data.

Evolutionary history of CRISPRs and cas genes in Salmonella enterica

Our results reveal that not all isolates have all cas genes and that two distinct cas gene profiles exist (Fig. 2), which raises some interesting evolutionary questions. For example, does the presence of some but not all cas genes within an isolate render the system non-functional? Furthermore, what is the evolutionary significance of having the CRISPR cassette of spacers and repeats but not having the cas genes as is the case with many isolates (Table S1)? The lack of a full set of cas genes observed here has also been observed elsewhere where a lack of function was assumed (e.g., within S. enterica subsp. arizonae and S. Paratyphi B Fricke et al., 2011). There are also examples within Escherichia coli of incomplete cas gene systems (Touchon & Rocha, 2010). However, generalizations about the functionality and fitness consequences of an incomplete set of cas genes are difficult as it may depend on the environment (Jiang et al., 2013). Additionally, recent studies have shown that at least in E. coli Cas1 and Cas2 are present in all fully functional CRISPR-Cas systems and that only those two genes and a single repeat are necessary for spacer integration (Yosef, Goren & Qimron, 2012). We found cas1 and cas2 had the smallest range in pairwise distances among the eight genes (Fig. S1), which may represent stronger purifying selection suggesting their importance to the functionality of the CRISPR-Cas system. Such a conclusion is in line with the results of Takeuchi et al. (2012), which found that cas1 and cas2 genes experience levels of purifying selection close to the genomic median but the other cas genes experienced much weaker purifying selection.

Among the isolates we investigated, there was no consistent pattern as to which cas genes were present when an isolate did not have all eight. For example, within many serovars (e.g., S. Abony, S. Chester, and S. Urbana) cas3 was the only cas gene present. Cas3 proteins have been proposed to be an important component of the CRISPR mechanism because they are involved in the cleavage of invading DNA (Beloglazova et al., 2011). Given this importance and the relatively high frequency of the pattern of only Cas3 being present among many isolates, perhaps those CRISPR systems with only Cas3 do serve some functional importance. Further studies are necessary to determine whether that is the case and what the evolutionary significance of having the CRISPR loci but none of the cas genes is. Because we used draft genomes, cas gene absence due to missing data cannot be ruled out.

Another interesting question that arises from our results is what is the evolutionary history and transmission mechanism responsible for the strong incongruence between a phylogeny based on cas genes and another based on a SNP matrix created from WGS data. A previous study of the CRISPR-Cas system within E. coli and Salmonella found that two distinct Salmonella clades existed based on variation within Cas1 proteins (Touchon & Rocha, 2010), which our results also confirm. However, what is difficult to interpret is that the two cas gene profiles are dispersed throughout the tree rather than clustered based on phylogeny and evolutionary history. This is true within the two general clades that have consistently been recovered through phylogenetic analyses (den Bakker et al., 2011b; Timme et al., 2013). As a result, it appears that there have been many non-vertical transmission events of large portions of the genome throughout the evolutionary history of S. enterica. A phylogeny based on the cas1 family across a very wide evolutionary breadth showed that CRISPR-Cas system subtypes are not reciprocally monophyletic and that there are instances of subtypes occurring in distant parts of the tree (Makarova, Wolf & Koonin, 2013). Earlier studies also implicated horizontal gene transfer in explaining the distribution of different CRISPR-Cas systems within closely related taxa. For example, one study discussed the role of megaplasmids as a vector for horizontally transferring the large region of DNA that constitutes the CRISPR-Cas system (Godde & Bickerton, 2006); another study noted that the presence of IS elements on both sides of the CRISPR-Cas system would likely facilitate horizontal transfer (Horvath et al., 2009). A CRISPR locus and the associated cas genes have also been found within a megaplasmid of the neutoroxigenic Clostridium butyricum Type E strains, which provides further evidence for the feasibility of horizontal transfer of the CRISPR-Cas system (Iacobino, Scalfaro & Franciosa, 2013).

Efficacy of CRISPRs for typing and subtyping

The increasingly sophisticated methods for assaying genomic DNA have resulted in novel markers for typing and subtyping bacterial isolates. These markers offer much more discriminatory power (e.g., ability to differentiate among closely related isolates) than the historical method of serotyping based on antigen profiles and the more recent method of PFGE (Allard et al., 2012; Underwood et al., 2013). The recent emphasis on the CRISPR system within Salmonella (Fabre et al., 2012; Shariat et al., 2013; Touchon & Rocha, 2010) is one such example where it represents a novel genetic element that may be suitable for typing. Within Salmonella this has either been illustrated, for example, through the sole use of the CRISPR loci (Fabre et al., 2012) or through the combination of CRISPR-MVLST and PFGE (Shariat et al., 2013). In practice, to use the CRISPR system for typing or subtyping requires the development of robust PCR primers and subsequent sizing via agarose gel electrophoresis or sequencing via capillary electrophoresis (Shi et al., 2013), both of which can be time consuming and require a non-trivial economic cost. Furthermore, PCR assays may not be universal such that they may be only serovar specific, which becomes an increasingly likely situation when developing high throughput assays necessary for daily surveillance like was the case with serovar Typhimurium (Fabre et al., 2012).

Given the costs of PCR based assays and the likely limited taxonomic breadth to which they can be applied, an important question is whether the performance of CRISPR loci for typing is good enough to overcome those issues. Our results suggest that CRISPR loci have some utility as a typing tool in that an appreciable number of isolates from the same serovar were clustered together (Table S2, Figs. 5–6). However, we found no significant differences in CRISPR diversity between outbreak and non-outbreak samples that were part of a published study on S. Montevideo (Allard et al., 2012), which is illustrated by the lack of differentiation into two groups of Montevideo in the DISTRUCT diagram (Fig. 7). The inability of CRISPR loci to differentiate among isolates associated with an outbreak was also observed within S. Agona where CRISPR spacer diversity was relatively constant across a diverse set of isolates spanning approximately 60 years of sampling and multiple outbreaks (Zhou et al., 2013). Additionally, the fact that there were many well supported lineages based on phylogenetic analyses of WGS data (Figs. 2–3 and Timme et al., 2013) and that methodology has been proven to discriminate among highly-clonal samples within outbreaks means that a more reliable alternative method to CRISPRs exists. We acknowledge that we have evaluated the utility of CRISPRs as a typing method using a novel approach and other methods analyzing such a system may not suffer the same error rate. However, those other methods also have drawbacks. For example, reference database approaches against which CRISPRs are queried to determine the serovar from which they came can be problematic if the database is poorly populated (e.g., Sorokin, Gelfand & Artamonova, 2010).

Whole genome sequencing and abandoning the target region paradigm

Given that PCR assays, of CRISPR loci or genomic targets in general, likely require a significant amount of development/validation and may only be serovar specific, perhaps it would be more efficient to perform whole genome sequencing and cluster isolates based on SNP differences. If feasible, it would appear such an approach offers the most information and does not suffer from many of the drawbacks associated with PCR assays. First, the discriminatory power provided by WGS data is superior to that provided by CRISPR loci, MLST, or PFGE. For example, we found that the clustering of isolates based on CRISPR spacer content or MLST sequence data had a higher error rate when compared to the SNP approach (Figs. 2–8; Table S1). Second, there are many instances where subtyping is inefficient to address the issues of concern. For example, within traceback investigations it is often a few SNPs (e.g., ten or so) that may differentiate outbreak from non-outbreak samples (Allard et al., 2012; Allard et al., 2013), which topologies based on CRISPR spacer content are unlikely to resolve such fine-scale relationships correctly. As a result, it is somewhat surprising that despite the decreasing cost of and increasing accessibility to NGS data there remains a focus on targeting a subset of the genome to type or subtype bacterial pathogens. However, the focus among international groups on the utility of next-generation sequencing and clustering based on whole genome sequencing (e.g., the Global Microbial Identifier and 100k Genomes initiatives) suggests that such an approach will eventually become the standard for typing and outbreak investigation.

Materials and Methods

Isolate sampling

We sampled 427 isolates which represented 64 different serovars within S. enterica ssp. enterica. We also included a single representative from four different subspecies for a total of 431 isolates (Table 1). A pure culture sample for each strain was taken from frozen stock, plated on Trypticase Soy Agar, and then incubated overnight at 37 °C. Cells were taken from the plate and inoculated into Trypticase Soy Broth culture for DNA extraction. All samples were representative cultures from a full-plate inoculation and were not single colonies. Genomic DNA was extracted using the Qiagen DNeasy kit (Qiagen, Valencia, CA, USA).

Genome sequencing, assembly and annotation

Of the 431 isolates we analyzed, 120 were newly sequenced as part of this study. The other 311 genomes were part of previous studies or genome announcements (see Table S3 for PubMed IDs and WGS accession numbers).

Four hundred and nine isolates were shotgun sequenced on the Roche 454 GS Titanium platform (Roche Diagnostics Corp., Indianapolis, IN, USA) (Table S3). Each isolate using this technology was run on a quarter of a titanium plate, which resulted in approximately 250,000 reads.

Twenty-one isolates were sequenced using Illumina’s MiSeq platform (Illumina, Inc., San Diego, CA, USA). Sample preparation and sequencing libraries were done using the Nextera Sample Preparation Kit; sequencing resulted in 2 X 151 bp paired in reads.

A single isolate (Table S3) was sequenced on the PacBioRS II (Pacific Biosciences, Menlo Park, CA, USA) and assembled using SMRT Analysis v2.0.1.

We created de novo assemblies for each isolate from the raw sequence data. The 454 reads were assembled using Roche’s Newbler Assembler v. 2.3–2.6 (Margulies et al., 2005) and Illumina reads were assembled with Ray v. 2.2.0 (Boisvert, Laviolette & Corbeil, 2010). Default parameters were used in all cases. Draft genomes of each sample (i.e., contigs) were annotated using NCBI’s Prokaryotic Genomes Automatic Annotation Pipeline (PGAAP). All downstream analyses were conducted using the annotated genomes. Although genome assembly of repetitive elements can be problematic, the fact that we were unable to isolate CRISPR or MLST loci from only a small number of samples (seven for CRISPR1 and one for CRISPR2) means that assembly issues did not significantly impact our results.

Whole genome phylogenetic analyses

We used the program kSNP (Gardner & Slezak, 2010) to construct a SNP matrix of the 431 samples to be used in downstream analyses. kSNP uses a k-mer approach to identify homologous single nucleotide polymorphisms among a group of individuals. Briefly, the program, through a series of Perl scripts, uses jellyfish (Marcais & Kingsford, 2011) to index all draft genomes into k-mers of length 25bp and SNPs are identified using MUMmer (Kurtz et al., 2004). We ran the analyses using the draft genomes for each isolate as input. For downstream analyses, we used a matrix within which each position had a nucleotide state for at least 90% of the samples, which is a compromise between a matrix of only core SNPs and a matrix of pan SNPs. We note that kSNP is likely to exclude repetitive and mobile elements from the SNP matrix as k-mers in those regions will either not be unique or will not be present in at least 90% of the samples, respectively.

A phylogenetic hypothesis of the evolutionary relationships among the 431 isolates was constructed using the approximately-maximum-likelihood inference method implemented in the program FastTreeMP v2.1.7 (Price, Dehal & Arkin, 2010). To assess topological support for relationships, we also constructed (due to computational constraints) 100 bootstrap replicates using the seqboot program within the phylip package (Felsenstein, 1989). Annotation of the non-bootstrapped tree with support values was performed using the Python library DendroPy (Sukumaran & Holder, 2010).

MLST Loci

Fasta sequences for each of the seven MLST loci (aroC, dnaN, hemD, hisD, purE, sucA, and thrA) described as useful for typing Salmonella (Achtman et al., 2012) were downloaded from http://mlst.ucc.ie/mlst/dbs/Senterica/Downloads_HTML. A BLAST (Altschul et al., 1997) database containing a representative of each MLST locus was then created against which the draft-genomes were BLASTed to isolate the MLST loci. Alignments were constructed using MUSCLE (Edgar, 2004) with default settings. Alignments were also manually curated to ensure the correct sequence was isolated and to merge into a single sequence instances where the MLST locus spanned multiple contigs. We constructed a phylogenetic hypothesis on the concatenated MLST alignment using the maximum-likelihood and genetic algorithm method employed in GARLI v2.0 (Genetic Algorithm for Rapid Likelihood Inference) (Zwickl, 2006). Analyses were performed using the grid computing resources associated with the Lattice Project (Bazinet et al., 2007) and the default settings within that service for GARLI analyses. To infer the best topology based on the observed data, we ran 100 replicate runs and present the tree with the best likelihood score. To assess topological support for relationships, we ran 1000 bootstrap replicates; annotation of the best tree with bootstrap support values was also performed using the Python library DendroPy (Sukumaran & Holder, 2010).

Taxonomic congruence within the whole genome, MLST, and CRISPR data

To measure how well the relationships depicted on the phylogenies constructed based on the SNP matrix deduced from the WGS data and based on the MLST data or CRISPR loci matched the expectations based on taxonomy (i.e., isolates from the same serovar should be monophyletic), we used the genealogical sorting index (gsi), which is a measure of genealogical exclusivity (Cummings, Neel & Shaw, 2008). The index ranges from 0 to 1, with the former representing a random arrangement of isolates on the trees with respect to their taxonomic identity and the latter represent complete exclusivity (reciprocal monophyly) under which all isolates belonging to the same serovar are clustered together. Analyses were based on calculating the weighted gsi statistic across 100 bootstrap replicates of each matrix (Timme et al., 2013). We note that under the current taxonomic alignment for the serovars we have sampled there are seven cases of polyphyly observed in another study (i.e., S. Agona, S. Bareilly, S. Kentucky, S. Muenchen, S. Newport, S. Paratyphi B, and S. Senftenberg, Timme et al., 2013). Under each of the four datasets, these serovars all had gsi values less than one and, therefore, these instances of polyphyly impacted the performance of the datasets equally.

To further evaluate the utility of the two datasets to differentiate serovars, we determined the degree of overlap in pairwise distances among isolates within and between serovar. Estimates of diversity were calculated using dna.dist within the R (R Development Core Team, 2011) package ape (Paradis, Claude & Strimmer, 2004).

Identification and analysis of cas genes and CRISPR loci

We identified the CRISPR associated (cas) genes using the following steps. First, we extracted all CDSs from 431 annotated draft genomes using a python script. We then used usearch (Edgar, 2010) to cluster the CDSs into orthologous gene clusters (using 85% sequence identity). Usearch commands:

usearch --sort allORFS.fasta --output sortorfs.fasta --maxlen 17000 usearch --cluster sortorfs.fasta --id 0.85 --uc results.uc usearch --uc2fastax results.uc --input sortorf.fasta -output results.fasta

Multi-gene fasta files were extracted using an in-house perl script. Cas gene clusters were identified by BLAST against the S. Typimurium LT2 reference genome (GenBank: AE006468; locus tags STM2937, STM2938, STM2939, STM2940, STM2941, STM2942, STM2943, STM2944). BLAST analyses were also conducted to determine whether a gene not found through our initial screening was actually absent or not detected due to poor annotation. If the gene was detected, we added it by hand. Each cas gene cluster was aligned using Muscle (Edgar, 2004) and then concatenated into one large multigene alignment comprising the seven cas genes. The cas gene ML phylogeny was reconstructed using RAxML (Stamatakis, 2006) (command: raxmlHPC -f a -s catCASgenes.phylip -x 12345 -#100 -m GTRCAT -n catCASgenes -N autoMRE -p 123).

We extracted Salmonella CRISPR loci 1 and 2 from each of the 431 draft genome assemblies. Each locus was analyzed separately where spacers and repeats were visualized with the CRISPR DB II Excel Macro (DuPont, Inc.; R Barrangou, 2012, unpublished data), as previously used (Horvath et al., 2008). Repeats were removed to determine the homology of spacers across strains, and the CRISPR spacer array was manually aligned to optimize the homology of spacers across Salmonella strains.

We used two approaches to determine how well the CRISPR loci could be used to type or subtype isolates. In both instances we evaluated the CRISPR loci independently and focused on spacer content. Within the first approach, we constructed a character matrix by extracting the spacer sequences, combining (not concatenating) them into a single file, and then using uclust (Edgar, 2010) to identify clusters of similar sequences (identity within each cluster set to 95%). Multiple values of percent identity were evaluated, which did not alter the results since the spacers are either highly similar or extremely different. Using this binary matrix, we then constructed a phenogram based on Euclidian distances and the UPGMA algorithm within the R package phangorn (Schliep, 2011). We also inferred topologies based on 1000 bootstrap replicates; replicate matrices were constructed using the ‘sample’ function within R. As with the SNP and MLST matrices, we also estimated pairwise intra- and inter-serovars for the CRISPR data.

The second approach employed a model-based Bayesian clustering method implemented in the program STRUCTURE (Falush, Stephens & Pritchard, 2003; Pritchard, Stephens & Donnelly, 2000). STRUCTURE assumes no a priori taxonomic membership but rather clusters individuals at a specific value of k (i.e., the number of clusters) based on similarities in a multi-character dataset. We used as input the presence/absence matrix based on spacers described above. For analyses with STRUCTURE, we only used serovars with >3 isolates. We used the default parameter settings, which included the model allowing for admixture. The analyses were run for values of k 2 through 20 for each CRISPR locus separately. We present the results for the best fitting value of k identified using Structure Harvester (Earl & Vonholdt, 2012), which utilizes the method described in Evanno et al. (Evanno, Regnaut & Goudet, 2005) that is based on changes in the likelihood score across different values of k. We also present the results fro the value of k corresponding to the actual number of serovars. Runs consisted of 70 000 generations with the first 20 000 serving as burnin.

Supplemental Information

Fig. S1 Pairwise distances within cas genes

Histograms of the proportional pairwise difference among isolates for each of the eight cas-associated genes.

Click here for additional data file.

Table S1 cas gene presence and absence within each isolate

Click here for additional data file.

Table S2 gsi across serovars for the different datasets analyzed

Different numbers of isolates per serovar across the datasets is due to the inability to always mine each genomic region from the whole genome sequence data.

Click here for additional data file.

Table S3 Taxonomy, assembly statistics, and accession information for the samples included in this study

Click here for additional data file.

We would like to thank C Wang, C Li and A Ottesen for generating draft Salmonella genomes. Wesley Morovic and Philippe Horvath at DuPont contributed helpful CRISPR sequence analyses, and the CRISPR macro, respectively.

Additional Information and Declarations

Competing Interests

Author Contributions

We declare no competing interests, financial, non-financial, professional, personal or otherwise.

James B. Pettengill conceived and designed the experiments, performed the experiments, analyzed the data, wrote the paper, prepared figures and/or tables, reviewed drafts of the paper.

Ruth E. Timme conceived and designed the experiments, performed the experiments, analyzed the data, wrote the paper, reviewed drafts of the paper.

Rodolphe Barrangou conceived and designed the experiments, performed the experiments, wrote the paper, reviewed drafts of the paper.

Magaly Toro performed the experiments.

Marc W. Allard reviewed drafts of the paper.

Errol Strain contributed reagents/materials/analysis tools, reviewed drafts of the paper.

Steven M. Musser and Eric W. Brown contributed reagents/materials/analysis tools.

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
