# Peer review of "The evolutionary history and diagnostic utility of the CRISPR-Cas system within Salmonella enterica ssp. enterica"

_PeerJ, doi:10.7717/peerj.340_

## Round 0.1 · original submission · Major Revisions

Your paper has been reviewed by two experts. While one has suggested only minor revisions, reviewer 2 has raised concerns about the design of your study. They have also raised the issue that the paper contains inaccuracies that suggest it has not been properly read by all the contributing authors.

Reviewer 1 ·

Basic reporting

No Comments

Experimental design

No Comments

Validity of the findings

No Comments

Additional comments

This manuscript describes comprehensive analyses of CRISPR-Cas systems in 432 Salmonella isolates representing different serovars. I think this is a novel interesting study; I only have minor suggestions that could make this manuscript better.

Line 35: change Senftenberg for “Salmonella Senftenberg”
Line 74: What it is known about CRISPR spacers matching Salmonella phage sequences?
Line 109 you used the term serotype and in line 116 you used serovar. I think you need to be consistent, use one term. WHO uses the term serovar.
Table 1: there is a number 10 in the line after Salmonella Eastbourne, this need to be fixed
Table 1: need a footnote explaining that length (L) refers to the number of spaces
Table 1: In the summary, 432 needs to be in the same line
Table 1: S. and enterica should be italicized
Figure 1: this figure needs a more descriptive legend.
Line 142: change Senftenburg for “Senftenberg”
Line 143: add an “s” to isolate
Figure 1: Indicate in the legend what cas type is in red and in blue.
Line 161: define gsi
Figure 3: MLST for S. Abony and CRISPR 1 for S. Bareilly have a white box, but white is not included in the color bar.
Line 245: change S. enterica for Salmonella enterica
Line 252: delete “?” after Table S1)
Line 265-266: these are results that are mentioned for the first time in the discussion. Move to the results section
Lines 276: delete “?”
Line 347: remove symbol “” after DNeasy
Line 355: remove symbol “” after Nextera
Discussion: I think a discussion of a practical example of how CRISPR spacers could be used for outbreak investigations would be good. Also a role of the environment in the diversity of CRISPRs spacers (e.g., isolates from farms with high phage prevalence or isolates from processing plants where phages are used as biocontrol?). Did you see more spacer diversity in clade A or clade B, or there is no difference? Did you see more spacer diversity in isolates with more Cas proteins? I think you could check if spacers match phage sequences.

Reviewer 2 ·

Basic reporting

This manuscript by Pettengill et al. is based on the analyses of the whole genome sequence data of 432 Salmonella enterica isolates that belonged to 69 different serovars resulting in a CRISPR sequence analysis method with the main conclusion that CRISPR based groupings are more accurate than MLST but not as accurate as the clustering from the whole genomic variation. I would suggest giving the authors a chance to extensively reanalyse the data and rewrite the manuscript more clearly for reconsideration.
Major basic issues:
1. The manuscript is badly written and many sentences are very confusing. For example, authors have used the terms ‘higher taxonomic level’ (line 63) and ‘lower taxonomic ranks’ (line 72) that are not defined and are unclear. Another example is the sentence ‘First, due to the evolutionary arms race between foreign elements (e.g., phages) and the host bacteria, the associated rapidly changing selection pressures cause CRISPRs to evolve quite quickly [9,20].’ (lines 77-79) clearly needs to be rephrased. Similar issues exist throughout the manuscript that needs re-writing.
2. Some statements are wrong and not supported by appropriate references. For example, ‘This is not surprising as Salmonella enterica subsp. enterica, which contains the Type IE CRISPR-Cas system [26], is the leading cause of bacterial food-borne disease in the United States [27,28].’ (lines 93-95). The authors failed to explain why they think that it is obvious for Salmonella enterica with Type IE system to be the major cause of food borne infection. They are citing Makarova et al. (2011) that is focussed on the evolution and classification of CRISPR-Cas systems and does not support this statement.
3. The authors have described Salmonella as a highly clonal organism (line 96) which is an entirely wrong concept. A series of publications from different groups from around the world have revealed the extensive recombination within Salmonella enterica. They should read Octavia and Lan (2006), Didelot et al. (2011) and Brown et al. (2003). It is surprizing that EW Brown, a senior author in this manuscript, is amongst the first to highlight widespread recombination in Salmonella.
4. The sentence ‘Although we had a heavily skewed sampling scheme where the vast majority of samples had less than three isolates, this does not account for the differences in array length where it appears that some serovars are more likely to acquire many unique spacers compared to others (e.g., Senftenburg and Muenster; Fig. 1)’ does not make sense and the conclusion that ‘some serovars are more likely to acquire many unique spacers compared to others’ is not supported.
5. Figures are poorly labelled with uninformative legends. For example, how can the Y-axis in Fig. 1 be CRISPR1 or CRISPR2? These probably are the numbers of spacers. Similarly, cas phylogeny in the Fig.2 lacked the scale and is not properly explained in the legend.

Minor issues:
1. Mistakes in the first lines of the abstract and introduction: CRISPR stands for clustered regularly interspaced short palindromic repeats and the word ‘interspersed’ should be replaced by interspaced.
2. Spacers have been defined twice in the introduction (lines 48-49 and line83).
3. The species names in references are not italicized.
4. Lines 115-117; ‘Timme et al. [14] also reported phylogenetic incongruence of CRISPR loci in Salmonella, although that study focused on between-serovar differences and did not investigate the potential subtyping capability of the CRISPR locus’. Phylogenetic incongruence of CRISPR loci with what?
5. Why not mention the exact numbers of strains sequenced using 454 and Illumina platform in methods (lines 349-355).
6. The methods needs to be more clearly described.

Experimental design

1. The authors are citing Achtman et al. (2012) for Salmonella MLST and completely overlooked the major finding that a serovar does not necessarily contains genetically related strains and many serovars included distinct groups. They should try to correlate the CRISPR diversity with the eBG groups as defined by Achtman et al. (2012) rather than serovars.
2. One of the major issues in the analyses of the whole genome sequences is the separation of the plasmid sequences. Many salmonella strains carry plasmids that can contaminate the genomic DNA extraction. The authors have not described how they avoided or removed the plasmid sequences. If they did not remove them, the whole genome phylogeny is not reliable as it will obscure the phylogenetic signal. Similarly they have used SNPs with missing alleles and did not mention anything about removing the SNPs in the mobile genetic elements including transposes, IS elements, repetitive sequences etc and some of these SNPs may have resulted due to misalignments of these elements.
3. It is unclear how the authors observed only 10,215 nucleotides in total for the MLST genes in 432 isolates? The total length of seven MLST genes is 3,336 bp and the 432 strains should make it 140,112 nucleotides.
4. Some results are interpreted wrong. For example, lines 176-178 where the authors blamed MLST for poor gsi values whereas these values are low because those serovars contain genetically unrelated isolates.
5. The major finding of this study is a novel method of CRISPR based grouping that is very confusing. How the authors grouped 432 isolates in 824 clusters from CRISPR 1 and 1,176 clusters from CRISPR2? How these findings improved our understanding of evolution and how they can be used for typing is neither properly mentioned nor appropriately discussed.
6. Another approach used was of STRUCTURE analysis and Fig. 7 shows the results of assuming 10 and 15 populations for both the CRISPR loci. Again why authors chose two different populations for STRUCTURE analyses and how the patterns correlate or differ to each other as well as to the groupings from the previous approach is unclear.

Validity of the findings

The overall conclusions of this study are not reliable due to poor concept and analysis approach as described above. It is known for long that the whole genome sequencing has greater discriminatory power than other typing schemes and is the most reliable tool for tracing the pathogen transmission and studying the evolution. Therefore, this study does not add anything new to what we already know.

Additional comments

I have several major issues with the concept and the design of the study and feel that it is not suitable for publication in the current form. This manuscript has authors including R. Barrangou who published great articles on CRISPR-Cas systems, yet CRISPR is described to stand for clustered regularly interspersed short palindromic repeats. Eric Brown was probably the first to show recombination in Salmonella in the PNAS publication in 2003, yet they described Salmonella to be highly clonal.

---

## Round 0.2 · Major Revisions

The reviewer noted that there are still several critical issues that were not addressed in your rebuttal and revised manuscript and that there are a number of discrepancies between the numbers and values throughout the manuscript. Please take care to address all the reviewers comments, and correct these discrepancies, in your revised manuscript.

Reviewer 1 ·

Basic reporting

The manuscript has improved after revision but not enough to be accepted for publication. I still have several concerns.

Experimental design

1. The authors previously mentioned that they have sequenced 432 strains belonging to 69 serovars. But different numbers are mentioned throughout the revised manuscript; sometimes 427 isolates from 64 serovars and sometimes 431 strains. According to methods, 431 isolates (427 from 64 different serovars and the remaining from four subspecies) were sequenced.
Now, authors previously observed 878 unique spacers plus 75 unique repeats within CRISPR1 and 1,241 unique spacers plus 65 unique repeats within CRISPR2 among 432 isolates of 69 serovars. The number of spacers and repeats remained unchanged for 427 isolates from 64 serovars and I suspect that these numbers are still incorrect because all the analyses were performed using 431 genomes.
2. The authors have also mentioned other studies where they have sequenced some genomes, e.g., Allard et al., 2012. It is not clear whether the total of 431 include these published genomes or this is entirely new set of strains which has not been analysed elsewhere.
3. The authors described two studies in the introduction (lines 102-110); one finding strong correlation of CRISPR diversity with serovars and the second describing them as poor epidemiological markers. This study lacks proper discussion and clear conclusion on whether CRIPSR based typing should be adopted for Salmonella enterica or not, in the light of the previous studies. They vaguely mentioned that ‘CRISPR loci have some utility as a typing tool in that an appreciable number of isolates from the same serovar were clustered together (Table S2, Fig. 5-6)’. In fact, the results suggest that CRISPR based typing is not reliable for Salmonella which does not properly correlate with serovars or the genetic groupings from the whole genomic variation and the MLST genes.
4. Language still has scope for improvements and the authors need to use proper evolutionary language where appropriate, for instance, ‘a high frequency of polyphyly’ should be ‘a high degree of polyphyly’ (line 66). The sentence in lines 64-68 needs rephrasing as ‘multiple instances of independent evolution [14], which is likely the result of horizontal transmission events’ does not make sense.
5. There are still wrong citations inserted into the text, for example, citations, 33, 34 and 35 in lines 96-97. The authors should find a suitable citation to support their statement on difficulties with the traditional genotyping methods in differentiating clonal populations.
6. The authors have claimed in their response to reviewers that no change in the conclusion from the MLST data was observed although they have previously used the wrong data due to an informatics error. However, they have made a significant change in their conclusion that MLST is more accurate than CRISPR typing whereas they previously concluded otherwise. The basis of this change is neither described in the manuscript nor described in the response to reviewers.
7. The result section ‘Intra- and inter-serovar pairwise distances’ is very badly written and the authors did not explain why the pairwise distance between Senftenberg and Tennessee was an exception. They talk about the similarities and differences in the pairwise distances between certain serovars but the values are not mentioned anywhere. Why use unnecessary fancy terms like DNA barcoding when they have simply performed inter- and intra- serovar pairwise distances.
8. In addition to the gsi values, it will be meaningful to describe the number of distinct phylogenetic groups within each serovars based on the genomic and MLST data, perhaps a table summarizing these groupings would be great. The manuscript will also benefit greatly with a discussion on the comparison of the gsi values with monophyletic and polyphyletic serovars based on the genomic, MLST and CRISPR diversity.
9. The authors concluded multiple horizontal transfer events for CRISPR-Cas acquisition despite they only formed two groups (lines 273-275). It is not clear why do the exclude the possibility that these systems were acquired once by the last common ancestor and then vertically transmitted.
10. It is not obvious why are the numbers of strains per serovars variable between the kSNP, MLST and CRISPR datasets for the calculation of gsi values (Table S2)?
11. The authors should avoid speculating strong purifying selection on cas1 and cas2 genes if they did not test the selection pressure (lines 154-155).
12. The authors have mentioned how they sequenced, assembled and annotated the genomes but no efforts were made to discuss the differences in the size of assemblies and coverage due to three different sequencing and assembling methods with the potential impact on the variation in overall diversity (including CRISPR diversity that may be affected due to repetitive nature of the sequences in the arrays).
13. The use of terms ‘higher taxonomic levels’, ‘lower taxonomic ranks’ and ‘putative taxonomy’ is avoidable that can be simplified for the precise phases like ‘across the bacterial species’ or ‘within each species’ or ‘within a serovar’ as appropriate.
14. The authors have tested the bootstrap support for the clustering pattern and talk about poor or good support in the results but no values are mentioned in any of the phylogenetic trees.
15. The bracket needs to be closed in line 173.
16. Why do they have table 2 if it is not cited anywhere in the text.
17. The genome accession numbers are not mentioned for all the strains in Table S3.
18. The species names in many references are still not italicized.

Validity of the findings

No Comments

---

## Round 0.3 · Minor Revisions

Your rebuttal was returned for review and the reviewer has requested some small revisions before your manuscript can be accepted.

Reviewer 1 ·

Basic reporting

See the general comments.

Experimental design

See the general comments.

Validity of the findings

See the general comments.

Additional comments

Manuscript can be accepted with the following minor changes.
1. The numbers of isolates in the abstract needs to be corrected as the values reflect the diversity at the CRISPR loci for 431 isolates.

2. For some reasons, the scale from the Cas-phylogeny in Fig.2 has disappeared again.

3. DNA barcoding is a concept that relies on the phylogenetic and distance-based analyses of particular genetic markers (short fragments of DNA) to identify different species and is still controversial. Pairwise nucleotide distance has been used long before the barcoding concept and is what has been applied in this study. I appreciate authors’ enthusiasm of applying new concepts to their research but they should simplify it as I previously suggested.

4. The statement in lines 196-199 “Focusing more on the clustering pattern of isolates from the same serovar, we found that the frequency of like-isolates being clustered together based on CRISPR1 or CRISPR2 was less than the results under the MLST or SNP dataset (under both CRISPR loci only 9 serovars had gsi values of 1; Table S2, Figs. 2 – 6)” needs to be re-written. “...like-isolates being clustered...” in the statement should be rewritten using more appropriate language, for example, “the clustering of isolates based on the similarities in the nucleotide sequences of ....”.

5. The statement in lines 211-222 needs correction. “.... individuals from different isolates appear ...”.

6. Lines 228-232: “There are also instances where intra-serovar pairwise distances are similar to the magnitude observed between serovars, which is primarily the result of isolates within Paratyphi B, Newport, and Kentucky, which is not surprising as these serovars are either polyphyletic or are monophyletic but have a great amount of genetic differentiation within them”. They are all polyphyletic according to MLST analyses (Achtman et al., 2012) and authors should clearly mention which of these serovars is monophyletic according to their analyses.

7. In line 277 “...the cleavage of invasive DNA...” invasive needs to be replaced by invading.

---

## Round 0.4 · accepted · Accept

We look forward to seeing your manuscript published.